# Rosiglitasone and ROCK Inhibitors Modulate Fibrogenetic Changes in TGF-β2 Treated Human Conjunctival Fibroblasts (HconF) in Different Manners

**DOI:** 10.3390/ijms22147335

**Published:** 2021-07-08

**Authors:** Yuika Oouchi, Megumi Watanabe, Yosuke Ida, Hiroshi Ohguro, Fumihito Hikage

**Affiliations:** Departments of Ophthalmology, School of Medicine, Sapporo Medical University, S1W17, Chuo-ku, Sapporo 060-8556, Japan; ecarlateaile@gmail.com (Y.O.); watanabe@sapmed.ac.jp (M.W.); funky.sonic@gmail.com (Y.I.); ooguro@sapmed.ac.jp (H.O.)

**Keywords:** TGFβ2, human conjunctival fibroblast, ROCK inhibitor, 3D culture, ripasudil, KD025, rosiglitazone

## Abstract

Purpose: The effects of Rho-associated coiled-coil containing protein kinase (ROCK) 1 and 2 inhibitor, ripasudil hydrochloride hydrate (Rip), ROCK2 inhibitor, KD025 or rosiglitazone (Rosi) on two-dimension (2D) and three-dimension (3D) cultured human conjunctival fibroblasts (HconF) treated by transforming growth factor (TGFβ2) were studied. Methods: Two-dimension and three-dimension cultured HconF were examined by transendothelial electrical resistance (TEER, 2D), size and stiffness (3D), and the expression of the extracellular matrix (ECM) including collagen1 (*COL1*), *COL4* and *COL6*, fibronectin *(FN*), and α-smooth muscle actin *(αSMA*) by quantitative PCR (2D, 3D) in the presence of Rip, KD025 or Rosi. Results: TGFβ2 caused a significant increase in (1) the TEER values (2D) which were greatly reduced by Rosi, (2) the stiffness of the 3D organoids which were substantially reduced by Rip or KD025, and (3) TGFβ2 induced a significant up-regulation of all ECMs, except for *COL6* (2D) or *αSMA* (3D), and down-regulation of *COL6* (2D). Rosi caused a significant up-regulation of *COL1, 4* and *6* (3D), and down-regulation of *COL6* (2D) and *αSMA* (3D). Most of these TGFβ2-induced expressions in the 2D and *αSMA* in the 3D were substantially inhibited by KD025, but *COL4* and *αSMA* in 2D were further enhanced by Rip. Conclusion: The findings reported herein indicate that TGFβ2 induces an increase in fibrogenetic changes on the plane and in the spatial space, and are inhibited by Rosi and ROCK inhibitors, respectively.

## 1. Introduction

It is well known that the human conjunctiva plays pivotal roles in serving as a physical protecting barrier and maintaining the homeostasis of the ocular surface [1]. Several diseases, as well as surgical intervention, can impair this type of conjunctival barrier function and can also lead to subconjunctival fibrosis. Among the alternative features of conjunctival barrier function, their permeability is also recognized to be an important factor for drug delivery to the posterior segment of the eye after the instillation of ocular drugs [2]. In terms of the clinical aspects of subconjunctival fibrosis, it has been suggested that the regulation of wound healing in the conjunctiva is of great importance in terms of the surgical outcomes of ocular surface-related diseases, such as pterygium and glaucoma [3,4,5,6,7]. In fact, conjunctival scarring at the operative site may adversely induce poor success rates in the subsequent trabeculectomy [8,9]. It is generally recognized that the fibroblast is the responsible cell in the normal wound healing process, as well as in the development of fibrosis [10]. It is well known that wound healing is a complex physiological response to an injury, and within this response, there are three main overlapping phenomena, i.e., inflammation, proliferation, and maturation [11]. In addition, it is also known that several cytokines and growth factors are involved in this wound healing process. Among these, transforming growth factor beta (TGF-β) regulates nearly all aspects of wound healing [11]. In fact, upon exposure to a variety of stimuli, particularly TGF-β, fibroblasts can be transdifferentiated into myofibroblasts [10,12,13], which are associated with smooth muscle cell characteristics and the expressed α-smooth muscle actin (α-SMA). Fibroblasts are recognized as one of the major sources of extracellular matrix (ECM) proteins, especially collagens (COLs), in addition to fibrogenic cytokines and chemokines. During the normal wound repair process, myofibroblasts undergo apoptosis and are removed from the wound area. Alternatively, if such a wound repairing process fails, progressive fibrosis by myofibroblasts may cause additional scar formation [12,14]. Therefore, preventing the conversion of fibroblasts to myofibroblasts and/or to decrease the production of ECM proteins by myofibroblasts and appropriate barrier function needs to be maintained to maintain healthy ocular surface conditions [8,15,16]. Interestingly, it has been demonstrated that the currently used anti-vascular endothelial growth factor (VEGF) agents, such as ranibizumab, bevacizumab and aflibercept, are frequently used in the treatment for VEGF-related diseases, and also cause TGF-β2 modulation [17]. Therefore, anti-VEGF agents have the potential for clinical use in regulating TGF-β2 expression in conjunctival fibroblasts.

The peroxisome proliferator-activated receptor γ (PPARγ), a nuclear receptor, binds with the retinoid X receptor (RXR) to form a dimer after ligation with its agonist. This heterodimer complex recognizes PPAR response elements (PPREs) within the promoters on target genes, and, in turn, regulates gene transcription for adipogenic differentiation [18,19,20]. Therefore, PPARγ agonists have specific promoting effects regarding the differentiation of fibroblasts into adipocytes [21,22]. Thus, PPARγ agonists have received great attention for their efficacy in regulating adipocyte differentiation as well as increasing insulin sensitivity in diabetic patients [20,23,24]. In fact, several synthetic PPARγ agonists, including thiazolidinediones (TZDs): trioglitazone, rosiglitazone, ciglitazone, and pioglitazone [23,24,25] (8, 25, 37), have been developed for the treatment of diabetes. Rosiglitazone (Rosi, Avandia, GlaxoSmithKline), one of the more specific TZDs for PPARγ [23,24], is currently used in the therapy of patients with type II diabetes. Alternatively, it was revealed that Rosi inhibits transforming growth factor (TGF)-β1-induced proliferation, migration, and myofibroblast differentiation of human Tenon’s fibroblasts (HTFs) by blocking the p38 signaling pathway [26].

It is known that Rho-associated coiled-coil containing protein kinase (ROCK) 1 and 2, serine/threonine kinases, are mainly involved in the regulation of cell movement through the formation of actin stress fibers and focal adhesions [27,28]. Both ROCK1 and ROCK2 can be activated in response to an injury and heparin binding–epidermal growth factor-like growth factor (HB-EGF) stimulation, and regulate cell–cell adhesion mediated by E-cadherin and β-catenin, in addition to the formation and maintenance of barrier integrity [29]. It was reported that the ROCK inhibitor (ROCK-i), Y-27632 or C3 facilitates wound healing by modulating cell–ECM and cell–cell adhesion or by wound closure, respectively, in corneal epithelium cells [29,30]. In one possible mechanism, ROCK-i inhibits the ROCK-induced up-regulation of TGF-β expression and the down-regulation of the expression of bone morphogenetic protein (BMP)-2, which mediates cell migration by inducing the formation of ECMs [31,32]. These above collective findings suggest that both Rosi and ROCK-i may be suitable candidates for changing TGF-β profibrotic signaling into pro-migratory BMP, resulting in the facilitation of the cell migration that is required for wound healing of conjunctiva. Furthermore, these observations also provide significant insights into other pathologies in which the TGF-β pathway and the role of the extracellular matrix is pivotal, such as cerebral cavernous malformations [33,34,35].

In our previous study, we reported on the development of a suitable in vivo model that replicates the glaucomatous human trabecular meshwork (HTM) by a three-dimension (3D) drop culture method [36,37,38] using TGF-β2-treated HTM cells in addition to the conventional two-dimension (2D) culture, and successfully evaluated the drug efficacy of ROCK-is on their TGF-β2-induced fibrogenetic changes [38]. Therefore, in the present study, to elucidate the effects of the stimulation of PPARγ by Rosi or the inhibition of ROCKs by pan-ROCK-i, ripasudil hydrochloride hydrate (Rip) or ROCK2-i, KD025 on TGF-β2-induced fibrogenetic changes of human conjunctival fibroblasts (HconF), 2D and 3D cultured HconF cells were used.

## 2. Results

### 2.1. Effects of the Stimulation of PPARγ or the Inhibition of ROCKs on TGF-β2-Treated 2D HconF Monolayers

In our recent study using TGF-β2-treated HTM cells replicating glaucomatous TM, we found that the TEER measurement data for the 2D HTM monolayer and a physical property analysis of the 3D sphenoids reflected different aspects of their TGF-β2-induced fibrogenetic changes [38]. In the present study, using this methodology, TGF-β2-induced fibrogenetic effects toward 2D cultured HconF cell monolayers were studied by TEER.

As shown in Figure 1, in the presence of a 5 ng/mL TGF-β2, the TEER value of 2D cultured HconF cell monolayers was significantly increased, while in contrast, the stimulation of PPARγ by rosiglitazone (Rosi) or the inhibition of ROCKs by pan-ROCK-i, Rip, or the selective ROCK2 inhibitor, KD025 did not induce any changes. Interestingly, these TGF-β2-induced increased TEER values were substantially inhibited by Rosi, but were not affected by Rip, or KD025. Thus, these data suggest that Rosi may inhibit the TGF-β2-induced increase in the fibrogenetic changes of the 2D cultured HconF monolayers.

### 2.2. Effects of the Stimulation of PPARγ or the Inhibition of ROCKs on the Physical Properties of the TGF-β2-Treated 3D HconF Sphenoids

In our recent study using 3D cultures of the HTM cell, we also found that TGF-β2 induced the formation of smaller and stiffer 3D sphenoids, which suggests that this 3D sphenoid culture may be useful for evaluating the spatial fibrogenetic changes [38]. Therefore, using a similar approach, to evaluate the spatial fibrogenetic effects of TGF-β2 of the 3D HconF sphenoids, their physical properties, including sizes and stiffness, were investigated in the absence or presence of Rip, KD025 or Rosi.

Surprisingly, as shown in Figure 2, the sizes of the sphenoidal 3D HconF sphenoids were significantly increased or decreased by both Rip and Rosi, although they were not affected by TGFβ2 nor KD025. In addition, sphenoids that were treated with either Rip, KD025 or Rosi were also not altered by the presence of TGFβ2.

Concerning the physical stiffness of the 3D HconF sphenoids, TGFβ2 and Rip induced significant increases or decreases, respectively, but Rosi or KD025 exerted no effect (Figure 3). However, in contrast to the sizes of the 3D sphenoids, the TGFβ2-induced increase in the stiffness was substantially inhibited by Rip or KD025, but not by Rosi. Since Rip and KD025 are inhibitors against ROCK1 and 2, and ROCK2, respectively, the above data indicate that the ROCK2 inhibition caused by TGFβ2 induced an increase in stiffness, and ROCK1 inhibition itself induced a decrease in stiffness. Therefore, these collective findings suggest that the TGFβ2-induced increase in physical stiffness of the 3D HconF sphenoids, which can be attributed to an increase in spatial fibrogenetic changes, were effectively suppressed by ROCK-is.

### 2.3. Effects of the Stimulation of PPARγ or the Inhibition of ROCKs on the mRNA Expression of TGF-β2-Treated 2D and 3D HconF Cells

To elucidate the mechanism responsible for the differences in the TEER values of 2D cultured HconF cells and the physical stiffness of the 3D HconF sphenoids among the above experimental groups, the expression of the major ECM associated with the 2D and 3D cultured HconF cells, including collagen1 (*COL1*), *COL4*, *COL6*, fibronectin (*FN*) and α-smooth muscle actin (*αSMA*), were evaluated by quantitative PCR (Figure 4). Upon administering TGFβ2 at a level of 5 ng/mL, the mRNA expression of *COL1*, *COL4*, *FN* and *α SMA*, or *COL6* of 2D cultured HconF cells were significantly up-regulated or down-regulated, respectively, while in contrast, in the 3D HconF sphenoids, TGFβ2 induced a marked up-regulation of *αSMA* expression. Interestingly, the Rosi-induced alteration of these ECM expressions were quite different between 2D and 3D cultured HconF cells, as follows: 2D cultured HconF cells: no significant changes except the down-regulation of *COL6*, but the TGFβ2-induced changes of *COL4* and *αSMA* were significantly inhibited; 3D HconF sphenoid: a significant up-regulation of *COL1, COL4* and *COL6*, and the down-regulation of *αSMA*, and the TGFβ2-induced changes of all ECMs were enhanced except for the suppression of αSMA. Similarly to Rosi, the effects of Rock-is on ECM expression in the absence or presence of TGFβ2 were also different between 2D and 3D cultured HconF cells; that is, no significant alteration was found in 3D sphenoids, but in 2D sphenoids, Rip caused the significant up-regulation of αSMA, and KD025 caused a substantial down-regulation of *COL6*. In terms of TGFβ2-induced changes, Rip significantly enhanced the expression of *COL4* and *αSMA* (2D) and suppression of *αSMA* (3D), and KD025 induced a marked suppression of all ECM molecules, except for *COL6* in 2D sphenoids, and *αSMA* in 3D sphenoids. These collective findings indicate that 2D cultured and 3D cultured HconF cells appear to reflect quite different aspects of the TGF-β2-induced fibrogenetic changes on the plane and in the spatial space, respectively.

## 3. Discussion

The 3D sphenoid culture method has attracted great interest as a possible model of specific diseases through an ex vivo approach [39]. In fact, a previous study suggested that, compared to the conventional 2D culture system, the 3D culture method can be more suitable for evaluating the structure of tissues, including the biological features that surround the network of ECM proteins [40]. However, some disadvantages have also pointed out, for example, difficulties in quantifying the target protein molecule because 3D sphenoids are composed of relatively smaller amounts of cells [39,41]. In our previous studies, we developed a 3D cell drop culture method that requires no scaffold-assist [36,37,42], and found that this ex vivo model could replicate the deepening upper eyelid sulcus (DUES) using 3T3-L1 cells and human orbital fibroblasts [36,37], or a glaucomatous trabecular meshwork (TM) using human TM cells [38]. In the current study, we also successfully produced uniform spheroidal 3D HconF sphenoids. In addition, since our 3D drop culture is a unique method for evaluating the physical stiffness of a single living 3D sphenoid by a micro-squeezer analysis, the TGF-β2-induced increase in their stiffness, which presumably replicates the TGF-β2-induced fibrogenetic changes of the 3D HconF sphenoids was also observed, similar to the 3D HTM sphenoids [38]. Furthermore, and interestingly, pan-ROCK-i, Rip, but not by the ROCK2 inhibitor, KD025, induced a substantial decrease in 3D sphenoid stiffness in addition to exerting significant enlargement effects. This result rationally suggests that ROCK1 but not ROCK2 may be related to the stiffness of the 3D HconF sphenoid. However, although ROCK2 inhibition by KD025 itself did not affect these physical properties of the 3D HconF sphenoid, KD025 effectively inhibited the TGFβ2-induced increase in their stiffness. Taken together, using this methodology, we successfully confirmed that the stiffness of 3D HconF sphenoids are significantly increased upon exposure to a TGFβ2 solution. This evidence rationally indicates that a TGFβ2-treated 3D HconF sphenoid may become a suitable ex vivo model for replicating conjunctival scaring. In contrast to the 3D cultured HconF cells, the findings of this study indicated that the conventional 2D cultures of HconF cells are still important in terms of evaluating fibrogenetic changes of the conjunctiva on the plane by TEER. Therefore, our current study suggests that 2D and 3D culture methods using HconF cells have the potential for use as a model for conjunctival fibrosis on the plane and in the spatial space, respectively.

It is well known that Rho-kinase (Rho-associated coiled-coil containing protein kinase; ROCK) is involved in a variety of physiological functions, including smooth muscle contraction, chemotaxis, and others [43,44]. The expression of ROCK isoforms, ROCK1 and ROCK2, is recognized to occur in various ocular tissues [45] and is also involved in the pathogenesis of several ocular diseases, including glaucoma, cataracts, corneal dysfunction, retinopathy and retinal dystrophy [46,47,48,49]. These observations, in turn, suggest that ROCK may represent a potential therapeutic target for the treatment of these diseases. In fact, ocular hypotensive effects caused by several ROCK inhibitors (ROCK-i) have been observed in several animal models [50,51], and one of the isoforms, ROCK-i, ripasudil hydrochloride hydrate (Rip), has recently been made available as a new member of an anti-glaucoma drop for the treatment of POAG and ocular hypertension [52]. Therapeutic possibilities for Rip will be also expected for the treatment of corneal endothelial dystrophy and inflammation of retinal pigment epithelium [53,54]. Furthermore, previous studies, using 2D cultured HconH cells [55] as well as human tenon fibroblasts [56], have demonstrated that Rip and other pan-ROCK-is also induce beneficial effects toward conjunctival fibrosis formation. Here, the findings reported herein reveal additional proof of the effects of ROCK-is, Rip and KD025 toward conjunctival fibrogenetic changes by a direct measuring stiffness of the 3D HconF cells.

Previous studies have revealed that TGF-β activates mitogen-activated protein kinase (MAPK) [57,58], and in turn, regulates the expression of collagen type I (*COL1*) expression in certain cells [59,60]. Such TGF-β-induced up-regulation in *COL1* expression was also recognized within HconF cells in the current study, as well as Tenon fibroblasts in a previous study [61]. Based upon the fact that PPAR-γ belongs to the nuclear hormone receptor superfamily of ligand-activated transcription factors, and has been found to regulate fibrosis in multiple organs [62,63], a previous study revealed that Rosi, a synthetic PPAR-γ agonist, could moderately antagonize TGF-β-induced fibrosis by blocking the TGF-β/Smad pathway in vitro [61]. In fact, Luo et al. demonstrated that Rosi can exert the anti-fibrotic activity of human tenon fibroblasts, thus suggesting that Rosi might be useful for modulating the scar formation that is frequently observed post-surgery [26,64]. Zhang et al. recently investigated the effects and toxicities of a poly (3-hydroxybutyric acid-co-3-hydroxyvaleric acid) (PHBV)-loading Rosi membrane toward scar formation after glaucoma filtration surgery (GFS) in a rabbit model and found following results: (1) the concentration of Rosi does not affect the morphology of the RSG/PHBV membrane, and (2) the RSG/PHBV membrane effectively and safely prevents the formation of fibrosis after GFS in a rabbit model. The implantation of the Rosi/PHBV membrane prevents scar formation after GFS. Therefore, they suggested that the Rosi/PHBV membrane would be a more effective and safer method than mitomycin C (MMC) for the prevention of GFS [65]. However, in contrast, our study using 2D and 3D cultured HconF cells demonstrated that Rosi significantly inhibited the TGFβ2-induced increase in TEER values, although Rosi did not induce significant effects on the TGFβ2-induced increase in the stiffness of the 3D Hcon F sphenoid as well as the up-regulation of ECM. Therefore, we propose that Rosi may be responsible for causing the inhibition of the TGF-β2-induced fibrogenetic changes on the plane, but not in the spatial space. In fact, such Rosi and other PPARγ agonist-induced suppressions of cell adhesion was also recognized in human colon cancer cells [66].

As there are study limitations of the current study, to better understand our present results related to PPARγ stimulation and ROCK inhibition toward TGF-β2-induced fibrogenetic changes of the 2D and 3D cultured HconF cells, further investigations including an RNA-Seq experiment to elucidate the linkages among PPARγ-related and ROCK-related signaling axes and several other mechanisms, such as oxidative stress, angiogenesis, or neurotransmission, will need to be examined.

## 4. Materials and Methods

### 4.1. Cell Culture and Treatment

Commercially available HconF cells (ScienCell Reserch laboratories, Carisbad, CA USA) were cultured in 150 mm 2D culture dishes until they reached 90% confluence at 37 °C in 2D medium composed of Fibroblast Medium (FM, Cat. #2301, ScienCell Reserch laboratories, Carisbad, CA, USA) and were maintained by changing the medium every other day. The 2D HconF cells were then further processed for transendothelial electron resistance (TEER) experiments or 3D sphenoid preparation, as described below.

The 3D HconF sphenoids were prepared following a recently published method for 3D cultures of human orbital fibroblasts (HOFs) [37,42] or human trabecular meshwork cells (HTM) [38]. Briefly, HconF cells at 90 % confluence as above were washed with phosphate buffered saline (PBS), detached using 0.25 % Trypsin/EDTA, and re-suspended in 3D medium composed of 2D medium supplemented with 0.25 % methylcellulose (Methocel A4M) after centrifugation for 5 min at 3000× *g*. Approximately 20,000 HconF cells in 28 μL of 3D medium were cultured in each well of the hanging drop culture plate (# HDP1385, Sigma-Aldrich, St Louis, MO, USA) (day 0). On each following day until day 6, half of the medium (14 μL) was replaced by fresh medium. For evaluation of the effects of drugs, in addition to 5 ng/mL TGFβ2, 50 µM ROCK-i, ripasudil hydrochloride hydrate (Rip, provided by the Kowa Company Ltd., Nagoya, Japan) or KD025 (Sigma-Aldrich, St Louis, MO, USA) or 10 µM rosiglitazone (Adipogen Life Science, Carisbad, CA, USA) was added to the 3D medium on days 1 through 6. The dosage of these TGFβ2, ROCK-is, or Rosi agents that were used were determined as the optimum dosage for our present study based on previous studies [55,64]. In terms of the concentrations of TGFβ2, several studies using human cell cultures indicated that the effects of the TGFβ were concentration-dependent, and that a dosage of approximately 5 ng/mL caused half maximum effects [67,68,69].

### 4.2. Transendothelial Electron Resistance (TEER) Measurements of 2D HTM Culture

The TEER values for HconF cell monolayers were determined by a previously described method [70]. Briefly, 2.0 × 10^4^ HconF cells prepared as above were seeded on each well of the TEER plate (0.4 μm pore size and 12 mm diameter; Corning Transwell, Sigma-Aldrich). In the absence or presence of 50 µM Rip, 50 μM KD025 or 10 μM rosiglitazone, 5 ng/mL TGFβ2-treated or untreated 2D HconF monolayers were cultured and the TEER values were measured at day 6.

### 4.3. Quantitative PCR

Total RNA extracted from 2D or 3D cultured HconF cells, reverse transcription and real-time PCR were processed as previously reported [36,37]. The sequences of primers and the Taqman probes used are shown in Table 1.

### 4.4. Solidity Measurements of 3D Organoids

The measurement of the physical solidity of 3D sphenoids using a micro-squeezer (MicroSquisher, CellScale, Waterloo, ON, Canada) was performed as described in a recent report [42]. A single sphenoid was placed on a 3 mm × 3 mm plate and compressed to 50 % deformation, as determined by a microscopic camera, for 20 s. The force required for the 50 % deformation of a single 3D sphenoid during 20 s was measured, and the force/displacement (μN/μm) then calculated.

### 4.5. Statistical Analysis

All statistical analyses were performed using Graph Pad Prism 8 (GraphPad Software, San Diego, CA, USA), as described recently [34,35].

## 5. Conclusions

Our newly developed 3D cell culture method permitted us to develop a better understanding of the molecular pharmacology of ROCK-is, Rip and KD025, or the PPAR agonist, Rosi toward TGF-β2 treated HconF cells, a possible model for affecting the conjunctival fibrogenetic changes on the plane and in the spatial space.

## Figures and Tables

**Figure 1 ijms-22-07335-f001:**
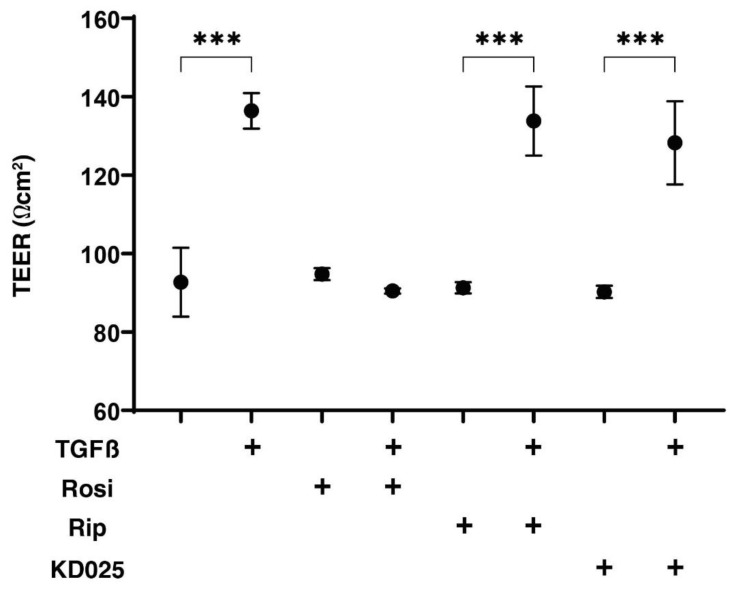
Transendothelial electrical resistance (TEER) of 2D cultures of HconF in the absence or presence of TGFβ2 and/or rosiglitazone, ripasudil or KD025.The following eight experimental groups included: (1) control for (2), (2) treated with 5 ng/mL TGF-β2 (TGFβ), (3) treated with 10 μM rosiglitazone (Rosi); control for (4), (4) treated with TGFβ and Rosi, (5) treated with 50 μM ripasudil (Rip); control for (6), (6) treated with TGFβ and Rip, (7) treated with 50 μM KD025; control for (8) and (8) treated with TGFβ and KD025. The electric resistance (Ωcm^2^) of 2D cultures of HconF monolayer at day 6 were evaluated by TEER measurement. All experiments were performed in triplicate using fresh preparations. Data are presented as the arithmetic mean ± the standard error of the mean (SEM). *** *p* < 0.005 (ANOVA followed by a Tukey’s multiple comparison test).

**Figure 2 ijms-22-07335-f002:**
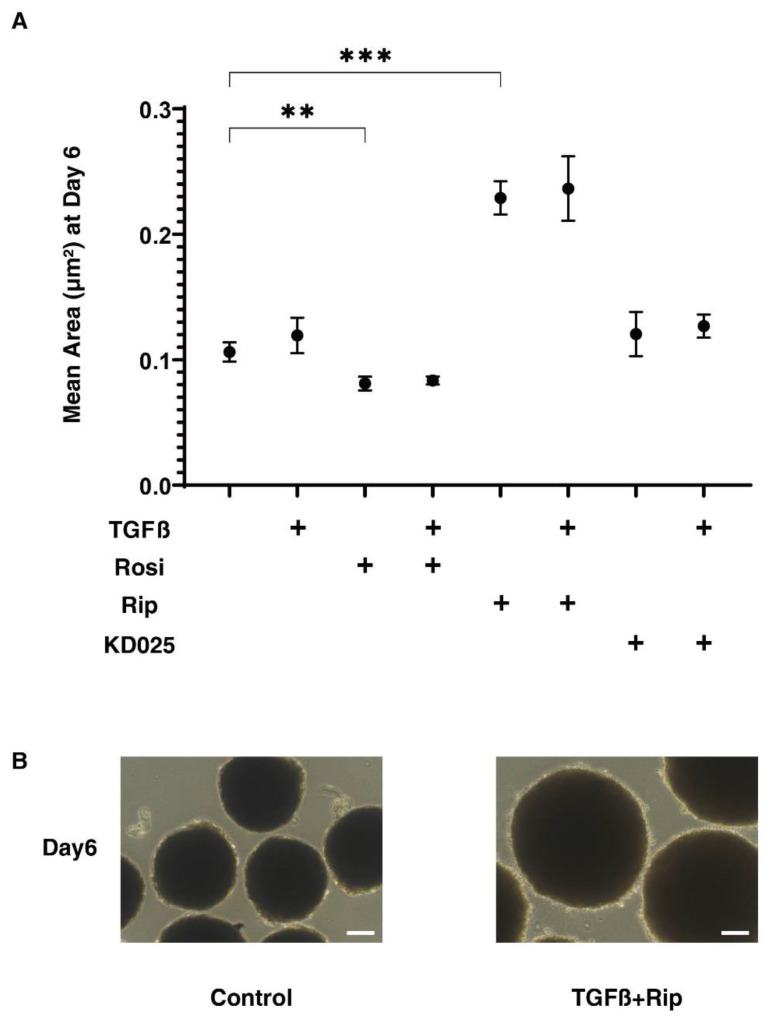
Mean sizes of 3D HconF organoids in the absence or presence of TGFβ2 and/or rosiglitazone, ripasudil or KD025.The following eight experimental groups included: (1) control for (2), (2) treated with 5 ng/mL TGF-β2 (TGFβ), (3) treated with 10 μM rosiglitazone (Rosi); control for (4), (4) treated with TGFβ and Rosi, (5) treated with 50 μM ripasudil (Rip); control for (6), (6) treated with TGFβ and Rip, (7) treated with 50 μM KD025; control for (8) and (8) treated with TGFβ and KD025.Mean sizes of 3D HconF organoids at Ddy 6 were measured and plotted (panel **A**). All experiments were performed in triplicate using fresh preparations consisting of 16 organoids each. Representative phase contrast images of control and TGFβ and Rip at day 6 are shown in (panel **B**) (scale bar: 100 μm). Data are presented as the arithmetic mean ± standard error of the mean (SEM). ** *p* < 0.01, *** *p* < 0.005 (ANOVA followed by a Tukey’s multiple comparison test).

**Figure 3 ijms-22-07335-f003:**
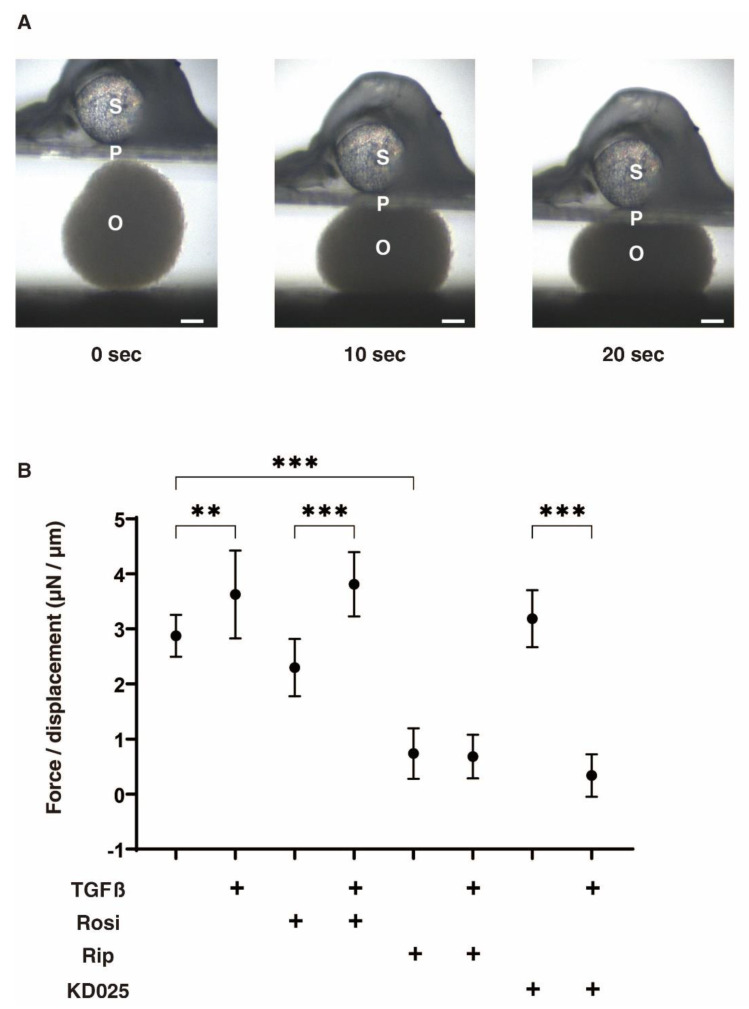
Physical stiffness of 3D HconF organoids in the absence or presence of TGFβ2 and/or rosiglitazone, ripasudil or KD025. The following eight experimental groups included: (1) control for (2), (2) treated with 5 ng/mL TGF-β2 (TGFβ), (3) treated with 10 μM rosiglitazone (Rosi); control for (4), (4) treated with TGFβ and Rosi, (5) treated with 50 μM ripasudil (Rip); control for (6), (6) treated with TGFβ and Rip, (7) treated with 50 μM KD025; control for (8) and (8) treated with TGFβ and KD025. Physical solidity of their 3D HconF organoids at day 6 were analyzed by a micro-squeezer (panel (**A**), S; pressure sensor, P; pressing plate, O; HconF organoid, scale bar: 100 µm), and requiring force to 50 % deformity of single organoid during 20 s were plotted in panel (**B**). All experiments were performed using 10 organoids freshly prepared each. ** *p* < 0.01, *** *p* < 0.005 (ANOVA followed by a Tukey’s multiple comparison test).

**Figure 4 ijms-22-07335-f004:**
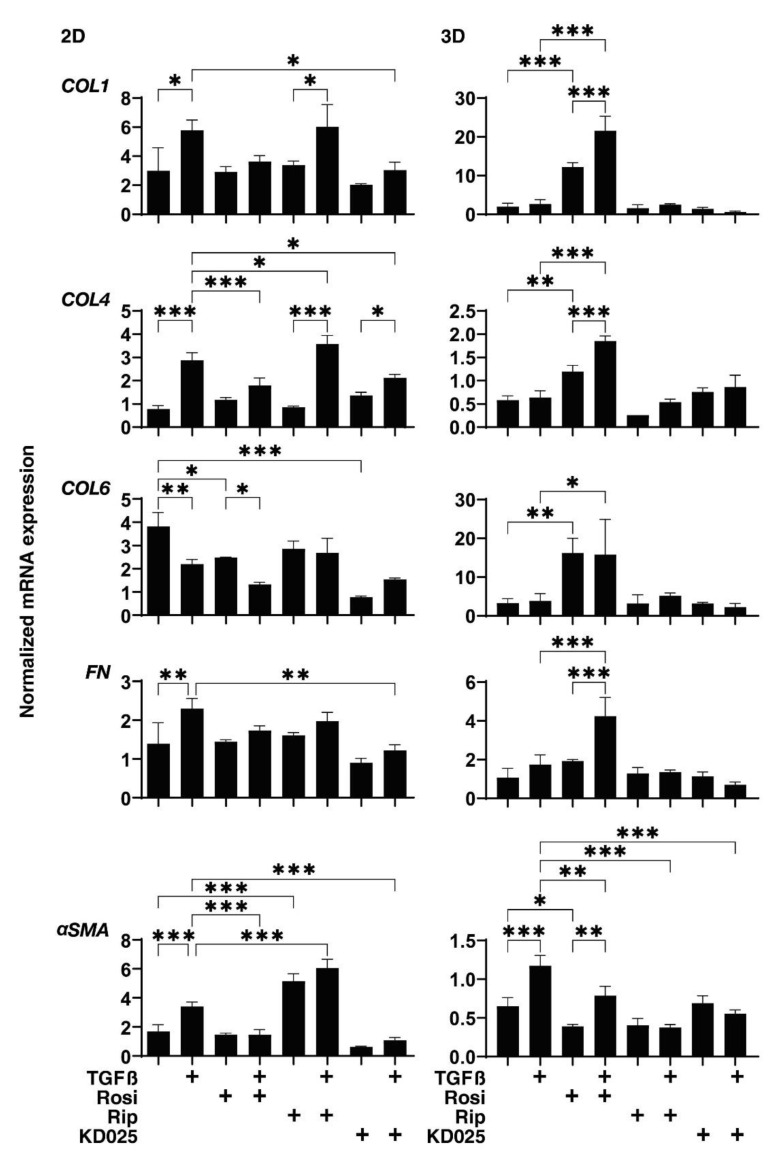
mRNA expression of ECMs of 2D and 3D cultured HconF cells in the absence or presence of TGFβ2 and/or rosiglitazone, ripasudil or KD025. The following eight experimental groups included: (1) control for (2), (2) treated with 5 ng/mL TGF-β2 (TGFβ), (3) treated with 10 μM rosiglitazone (Rosi); control for (4), (4) treated with TGFβ and Rosi, (5) treated with 50 μM ripasudil (Rip); control for (6), (6) treated with TGFβ and Rip, (7) treated with 50 μM KD025; control for (8) and (8) treated with TGFβ and KD025. Each 2D and 3D HconF cells at day 6 was subjected to qPCR analysis to estimate the expression of mRNA in ECMs (*COL1, COL4, COL6* or *FN*). All experiments were performed in duplicate using fresh preparations consisting of a single well out of 12 wells of the culture dish (2D) or 16 organoids each. Data are presented as the arithmetic mean ± the standard error of the mean (SEM). * *p* < 0.05,** *p* < 0.01, *** *p* < 0.005 (ANOVA followed by a Tukey’s multiple comparison test).

**Table 1 ijms-22-07335-t001:** The sequences of primers and the Taqman probes.

		Sequence	Exon Location	RefSeqNumber
human RPLP0	Probe	5′-/56-FAM/CCCTGTCTT/ZEN/CCCTGGGCATCAC/3IABkFQ/-3′	2–3	NM_001002
	Primer2	5′-TCGTCTTTAAACCCTGCGTG-3′		
	Primer1	5′-TGTCTGCTCCCACAATGAAAC-3′		
human COL1A1	Probe	5′-/56-FAM/TCGAGGGCC/ZEN/AAGACGAAGACATC/3IABkFQ/-3′	1–2	NM_000088
	Primer2	5′-GACATGTTCAGCTTTGTGGAC-3′		
	Primer1	5′-TTCTGTACGCAGGTGATTGG-3′		
human COL4A1	Probe	5′-/56-FAM/TCATACAGA/ZEN/CTTGGCAGCGGCT/3IABkFQ/-3′	51–52	NM_001845
	Primer2	5′-AGAGAGGAGCGAGATGTTCA-3′		
	Primer1	5′-TGAGTCAGGCTTCATTATGTTCT-3′		
human COL6A1	Primer2	5′-CCTCGTGGACAAAGTCAAGT-3′	2–3	NM_001848
	Primer1	5′-GTGAGGCCTTGGATGATCTC-3′		
human FN1	Primer2	5′-CGTCCTAAAGACTCCATGATCTG-3′	3–4	NM_212482
	Primer1	5′-ACCAATCTTGTAGGACTGACC-3′		
human αSMA	Probe	5′-/56-FAM/AGACCCTGT/ZEN/TCCAGCCATCCTTC/3IABkFQ/-3′	8–9	NM_001613
	Primer2	5′-AGAGTTACGAGTTGCCTGATG-3′		
	Primer1	5′-CTGTTGTAGGTGGTTTCATGGA-3′		

## Data Availability

The data that support the findings of this study are available from the corresponding author upon reasonable request.

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
