# Peer review of "Rosiglitasone and ROCK Inhibitors Modulate Fibrogenetic Changes in TGF-β2 Treated Human Conjunctival Fibroblasts (HconF) in Different Manners"

_ijms, 2021, doi:10.3390/ijms22147335_

Round 1
Reviewer 1 Report
Oouchi et al. produced a very well-written article focused on how “Treatment of 2D and 3 D cultured human conjunctival fibroblasts (HconF) with transforming growth factor (TGF)-β2 causes the deterioration of the surface barrier and subepithelial fibrosis”. I consider the manuscript very fascinating but, in the same time, I suggest several revisions needed to improve the reliability and the completeness of the paper:
- “Introduction”, lines 79-82. It could be interesting to apply the realized study on Rosi and ROCK-i to other pathologies in which the TGF-beta pathway and the role of extracellular matrix is pivotal, such as cerebral cavernous malformations. Regarding this, I suggest the authors to cite interesting bibliography on CCM proteins, like PMID: 27737651, PMID: 28870584 and PMID: 26115622.
- “Materials and Methods”: did the authors realize all experiments at least in triplicate?
- Section 4.3, “Quantitative PCR”: I suggest the authors to shift the primer list to a clearer table rather than place it through the main text.
- “Discussion” section, lines 219-231. The Rho kinase axis is already explored in relationship to retinal dystrophies. Thus, as future perspective, I suggest the authors to add the possibility to realize an RNA-Seq experiment, in order to enforce results obtained in this work, especially improving the link between the Rho axis and several other ones, such as oxidative stress, angiogenesis, or neurotransmission. Several papers already published could be used as reference, involving also inflammatory-related diseases. Regarding these, I suggest to add the following references to manuscript PMID: 33233726, PMID: 33233546 and PMID: 33374679.
- Finally, manuscript requires English revisions and typos correction.
Author Response
Dear Editor,
Thank you very much for the constructive comments concerning our revised manuscript entitled “Treatment of 2D and 3 D cultured human conjunctival fibroblasts (HconF) with transforming growth factor (TGF)-β2 causes the deterioration of the surface barrier and subepithelial fibrosis“. We carefully checked all of the additional comments made by the reviewers and have made a series of specific changes to our manuscript as follows;
Reviewer Comments:
The paper entitled “Treatment of 2D and 3 D cultured human conjunctival fibroblasts (HconF) with transforming growth factor (TGF)-β2 causes the deterioration of the surface barrier and subepithelial fibrosis” by Yuika Oouchi deals with the study of the role of (TGF)-β2 on human conjunctival fibroblasts.
Although the paper addresses an issue of interest in the field, the authors may wish to consider the following prior to publication.
Points
- The authors used only one concentration of 10 μM ROCK-i, ripasudil, KD025, 10 μM rosiglitazone. Why? The should provide strong rationale to use only one concentration. Incidentally, also the 5 ng/mL TGFβ2 needs full explanation.
Answer; Thank you for this comment. First, I deeply apologize that we made an error in the concentration of ROCK inhibitor used (10 μM). We carefully reviewed our experimental notebook and the correct concentration is 50 μM, and this concentration was based on previous work by Tanihara and his asscociates (Experimental Eye Research 2016, 149: 107-115. The effects of ripasudil (K-115), a Rho kinase inhibitor, on activation of human conjunctival fibroblast, ref # 36). In this study, they tried several concentrations of ripasudil (0, 25, 50, 75 and 100 μM) and 50 μM is the optimum concentration preventing the TGFb2 (5 ng/ml) induced effects toward HconF cells. In terms of the concentration of rosiglitazone, this concentration was also based on a previous in vivo study of TGFb2 treated rabbit conjunctiva (Zhang F et al. Rosiglitazone Treatment Prevents Postoperative Fibrosis in a Rabbit Model of Glaucoma Filtration Surgery. Invest Ophthalmol Vis Sci. 2019 Jun 3;60(7):2743-2752. doi: 10.1167/iovs.18-26526. Ref # 37). In terms of the concentration of TGFb2, 1-20 ng/ml TGFb concentrations were used in several cell culture experiments using human cells as follows; 1) human keratinocyte: 1-10 ng/ml TGFb (Sarret Y et al. Human keratinocyte locomotion: the effect of selected cytokines. J Invest Dermatol. 1992 Jan;98(1):12-6. doi: 10.1111/1523-1747.ep12493517.), 2) Human trabecular meshwork cells: 1-10 ng/ml TGFb (Fitzgerald AM et al. The Effects of Transforming Growth Factor-β2 on the Expression of Follistatin and Activin A in Normal and Glaucomatous Human Trabecular Meshwork Cells and Tissues. Invest Ophthalmol Vis Sci. 2012 Oct 23;53(11):7358-69.), 3) Human pigment epithelium cells: 1-20 ng/ml TGFb (Bian Z-M et al. Regulation of VEGF mRNA Expression and Protein Secretion by TGF-β2 in Human Retinal Pigment Epithelial Cells. Exp Eye Res. 2007 May;84(5):812-22. doi: 10.1016/j.exer.2006.12.016. Epub 2007 Jan 9). These studies indicated that the effects of the TGFb were concentration dependent and approximately 5 ng/ml caused half maximum effects. Thus, most of the study used a concentration of 5 ng/ml. In addition, in the previous work by Tanihara and his asscociates (Experimental Eye Research 2016, 149: 107-115. The effects of ripasudil (K-115), a Rho kinase inhibitor, on the activation of human conjunctival fibroblast, ref # 36), a concentration of 5 ng/ml was used. Therefore, the above information are included in the method section; “Commercially available HconF cells (ScienCell Reserch laboratories, CA USA) were cultured in 150 mm 2D culture dishes until reaching 90 % confluence at 37°C in 2D medium composed of Fibroblast Medium (FM, Cat. #2301, ScienCell Reserch laboratories, CA USA) and were maintained by changing the medium every other day. The 2D HconF cells were then further processed for transendothelial electron resistance (TEER) experiments or 3D sphenoid preparation as described below. The 3D HconF sphenoids were prepared following a recently published method for 3D cultures of human orbital fibroblasts (HOFs) [33,35] or human trabecular meshwork cells (HTM) [37]. Briefly, HconF cells at 90 % confluence as above were washed with phosphate buffered saline (PBS), detached using 0.25 % Trypsin/EDTA, and re-suspended in 3D medium composed of 2D medium supplemented with 0.25 % methylcellulose (Methocel A4M) after centrifugation for 5 min at 300 x g. Approximately, 20,000 HconF cells in 28 μL of 3D medium were cultured in each well of the hanging drop culture plate (# HDP1385, Sigma-Aldrich) (Day 0). On each following day until Day 6, half of the medium (14 ml) was replaced by fresh medium. For evaluation of the effects of drugs, in addition to 5 ng/mL TGFβ2, 50 µM ROCK-i, ripasudil (Rip, provided by the Kowa Company Ltd., Nagoya, Japan) or KD025 (Sigma-Aldrich, St Louis, MO) or 10 µM rosiglitazone (Adipogen Life Science, CA USA) was added to the 3D medium on Days 1 through 6. The dosage of these TGFβ2, ROCK-is, or Rosi agents that are used were determined as the optimum dosage for our present study based on previous studies [38,39]. In terms of the concentrations of TGFβ2, several studies using human cell cultures indicated that the effects of the TGFb were concentration dependent and that a dosage of approximately 5 ng/ml caused half maximum effects [40,41][42].”
Reviewer 2 Report
In this study, it seems that the authors investigated the effects of PPARγ stimulation or ROCK inhibition on TGF-β2-treated 2D and 3D cultured human conjunctival fibroblasts. There are inconsistencies in the main theme of the paper, which leads to difficulty in determining the value of the manuscript. Major and Minor concerns are as follows.
Major concerns
1. Title: The authors failed to demonstrate 1) the deterioration of the surface barrier and 2) subepithelial fibrosis. They only measured TEER, which cannot be directly linked to beneficial or detrimental effects of barrier function by itself. They did not show "subepithelial" fibrosis. Furthermore, it is not clear whether the authors intended to investigate the effects of TGF-β2 or PPARγ stimulation or ROCK inhibition on TGF-β2-treated systems.
2. Introduction: It is not clear why the authors investigated the effects of PPARγ stimulation or ROCK inhibition on TGF-β2-treated 2D and 3D cultured human conjunctival fibroblasts.
3. Please provide further data showing that increases in TEER are linked to the deterioration of the barrier, such as the change of molecules related to barrier function according to the experimental groups.
4. Figure 1-3: What did the authors intend to show? The effects of TGF-β2? PPARγ stimulation or ROCK inhibition without TGF-β2? Or PPARγ stimulation or ROCK inhibition on TGF-β2?
5. Please provide discussion or further data on differential effects of TGF-β2, PPARγ stimulation, or ROCK inhibition on studied aspects such as TEER and organoid size.
6. Please provide further data on the mechanisms affecting barrier function and fibrosis.
7. Please indicate specific statistical methods in each figure legend. It is not clear in which experiments the authors performed Student's t-tests.
Minor concerns
1. There are several typos, spacing errors, and misuse of abbreviations. Please also revise the manuscript thoroughly in terms of grammar.
2. It is inappropriate to use the brand name of drugs in the manuscript.
3. It is recommended to add the titles of subsections in the Results section.
4. Line 100 ends abruptly.
Author Response
Dear Editor,
Thank you very much for the constructive comments concerning our revised manuscript entitled “Treatment of 2D and 3 D cultured human conjunctival fibroblasts (HconF) with transforming growth factor (TGF)-β2 causes the deterioration of the surface barrier and subepithelial fibrosis“. We carefully checked all of the additional comments made by the reviewers and have made a series of specific changes to our manuscript as follows;
Reviewer Comments:
The paper entitled “Treatment of 2D and 3 D cultured human conjunctival fibroblasts (HconF) with transforming growth factor (TGF)-β2 causes the deterioration of the surface barrier and subepithelial fibrosis” by Yuika Oouchi deals with the study of the role of (TGF)-β2 on human conjunctival fibroblasts.
Although the paper addresses an issue of interest in the field, the authors may wish to consider the following prior to publication.
- The authors focused their study on TGFβ2, even the title underline that; it should be useful to emphasize TGFβ2 in the Introduction Section adding a brief introduction on TGFβ2. Further, to provide a wide perspective to the reader I suggest to add the following in the Introduction Section: “It has been demonstrated in conjunctival fibroblasts that anti-VEGF agents can regulate the TGF-β2 expression. On this regards it is worth of note that anti-VEGF agents, used in clinical practice, such as ranibizumab, bevacizumab and aflibercept are considerably different in terms of molecular interactions when they bind VEGF with and impact also on TGF-β2 modulation (Platania et al. Front Pharmacol. 2015 Oct 29;6:248. doi: 10.3389/fphar.2015.00248); therefore, characterization of such features can improve the design of novel biological drugs potentially useful in clinical practice.” Reference section: please add this relevant paper: Platania CB, Di Paola L, Leggio GM, Romano GL, Drago F, Salomone S, Bucolo C. Molecular features of interaction between VEGFA and anti-angiogenic drugs used in retinal diseases: a computational approach.Front Pharmacol. 2015 Oct 29;6:248. doi: 10.3389/fphar.2015.00248
Answer; Thank you very much for this constructive comment. As suggested a brief introduction for TGFb and information regarding anti-VEGF agents are now included in the first paragraph of the introduction; “It is well known that the human conjunctiva plays pivotal roles in serving as a physical protecting barrier and maintaining homeostasis of the ocular surface [1]. Several diseases as well as surgical intervention can impair this type of conjunctival barrier function and can also lead subconjunctival fibrosis. Among the alternative features of conjunctival barrier function, their permeability is also recognized to be an important factor for drug delivery to the posterior segment of the eye after the instillation of ocular drugs [2]. In terms of the clinical aspects of subconjunctival fibrosis, it has been suggested that the regulation of the wound healing in the conjunctiva is of great importance in terms of the surgical outcomes of ocular surface related diseases, such as pterygium, glaucoma [3-7]. In fact, conjunctival scarring at the operative site may adversely induce poor success rates in the subsequent trabeculectomy [8,9]. It is generally recognized that the fibroblast is the responsible cell in the normal wound-healing process, as well as in the development of fibrosis[10]. It is well known that wound healing is a complex physiological response to an injury and within this response, there are three main overlapping phenomena, i.e., inflammation, proliferation, and maturation [11]. In addition, it is also known that several cytokines and growth factors are involved in this wound healing process. Among these, transforming growth factor beta (TGF-β) regulates nearly all aspects of wound healing [11]. In fact, upon exposure to a variety of stimuli, particularly TGF-β, fibroblasts can be transdifferentiated into myofibroblasts [10,12,13], which are associated with smooth muscle cell characteristics and the expressed α-smooth muscle actin (α-SMA). Fibroblasts are recognized as one of the major sources of extracellular matrix (ECM) proteins, especially collagens (COLs), in addition to fibrogenic cytokines and chemokines. During the normal wound repair process, myofibroblasts undergo apoptosis and are removed from the wound area. Alternatively, if such wound repairing process fails, progressive fibrosis by myofibroblasts may cause additional scar formation[12,14]. Therefore, preventing the conversion of fibroblasts to myofibroblasts and/or to decrease the production of ECM proteins by myofibroblasts and appropriate barrier function needs to be maintained to maintain healthy ocular surface conditions [8,15,16]. Interestingly, since it has been demonstrated that anti-VEGF agents, which are used in clinical practice, such as ranibizumab, bevacizumab and aflibercept are frequently used in the treatment for VEGF-related diseases, and also cause TGFβ2 modulation[17]. Therefore, anti-VEGF agents have the potential for use in clinical practice to regulate TGF-β2 expression in conjunctival fibroblasts.”.
Round 2
Reviewer 1 Report
The manuscript can be accepted in the present form.
Author Response

(The authors gave the same response as above.)

Reviewer 2 Report
I do not know why, but nothing has been resolved in regard to my concerns.
Round 3
Reviewer 2 Report
Unfortunately, the authors did not properly address the previous concerns. It is hard to follow what the authors intended to suggest in the manuscript. They did not perform the analyses or show the results to demonstrate the argued conclusions.
Major points
The authors did not provide the data demonstrating the changes of molecules related to TEER change.
Also, they failed to provide the analyses to demonstrate the effects of drugs on TGF-β2-treated fibroblasts.
Minor points
'sphenoids' might be 'spheroids'